# Evaluating interhemispheric connectivity during midline object recognition using EEG

**Anwesha Das**[1], **Alexandra Mandel**[2,3], **Hitoshi Shitara**[2,4], **Traian Popa**[2], **Silvina G. Horovitz**[2], **Mark Hallett**[2], **Nivethida Thirugnanasambandam**[1,2]*

**1** Human Motor Neurophysiology and Neuromodulation Lab, National Brain Research Centre (NBRC), Manesar, Haryana, India, **2** Human Motor Control Section, National Institute of Neurological Disorders and Stroke (NINDS), National Institutes of Health, Bethesda, Maryland, United States of America, **3** The George Washington University, Washington, DC, United States of America, **4** Department of Orthopaedic Surgery, Gunma University Graduate School of Medicine, Tokyo, Japan

* dr.nivethida@gmail.com, nivethida@nbrc.ac.in

**Data Availability Statement:** The data have been made available at the following link: https://data.ninds.nih.gov/Hallett/NThirugnanasambandam/Das_et_al/index.html.

## Abstract

Functional integration between two hemispheres is crucial for perceptual binding to occur when visual stimuli are presented in the midline of the visual field. Mima and colleagues (2001) showed using EEG that midline object recognition was associated with task-related decrease in alpha band power (alpha desynchronisation) and a transient increase in interhemispheric coherence. Our objective in the current study was to replicate the results of Mima et al. and to further evaluate interhemispheric effective connectivity during midline object recognition in source space. We recruited 11 healthy adult volunteers and recorded EEG from 64 channels while they performed a midline object recognition task. Task-related power and coherence were estimated in sensor and source spaces. Further, effective connectivity was evaluated using Granger causality. While we were able to replicate the alpha desynchronisation associated with midline object recognition, we could not replicate the coherence results of Mima et al. The data-driven approach that we employed in our study localised the source of alpha desynchronisation over the left occipito-temporal region. In the alpha band, we further observed significant increase in imaginary part of coherency between bilateral occipito-temporal regions during object recognition. Finally, Granger causality analysis between the left and right occipito-temporal regions provided an insight that even though there is bidirectional interaction, the left occipito-temporal region may be crucial for integrating the information necessary for object recognition. The significance of the current study lies in using high-density EEG and applying more appropriate and robust measures of connectivity as well as statistical analysis to validate and enhance our current knowledge on the neural basis of midline object recognition.

## Introduction

Visual object recognition involves complex information processing and coordinated communication between several brain regions across the two hemispheres. When an object stimulus is

**Funding:** The work was supported by NINDS intramural research program. NT was funded by Ruth L. Kirschtein NRSA award from NINDS (2013-2016) and DBT/WT India Alliance fellowship grant IA/CPHI/16/1/502624 (2017-present). AD was funded by the Master's research fellowship from the Department of Biotechnology, Ministry of Science and Technology, India. The funders had no role in study design, data collection and analysis, decision to publish, or preparation of the manuscript.

**Competing interests:** The authors have declared that no competing interests exist.

presented at the centre of the visual field, the image from the right hemifield is formed on the left visual cortex and that from the left hemifield is formed on the right visual cortex. Although our sensory systems use a divide-and-conquer strategy, our perception is of a unified whole [1]. Therefore, for the brain to perceive a complete image of the object, the two internal images need to be integrated. This phenomenon is called perceptual binding [2–5]. For perceptual binding to occur, there has to be some form of communication or functional integration between the two hemispheres. One of the widely accepted views on neuronal communication is the 'communication through coherence (CTC)' hypothesis [6–9]. It proposes that distinctly located neuronal groups can communicate with one another by virtue of phase-locked neuronal oscillations known as neuronal coherence. The communication between the two hemispheres during object recognition may also occur via such neuronal synchronisation or coherence. Researchers in the past have studied these neuronal synchronisations using EEG [10, 11]. A study by Mima and colleagues [12] identified transient interhemispheric occipito-temporal synchronisation of alpha rhythm (frequency range 8–12 Hz) during a midline object recognition task using 29-channel EEG. Their study demonstrated a significant increase in interhemispheric coherence in the 117–373 ms time window when healthy human participants viewed familiar object stimuli compared to when they viewed non-familiar or meaningless object stimuli. They also observed that in the later time window of 373–639 ms, there was a significant decrease in alpha power for both object and meaningless object stimuli that were displayed in the midline of the visual field. However, the study was not without limitations. The authors presumed that the power and coherence changes would be best observed over a predetermined set of 4 electrodes that were positioned over the occipito-temporal regions. We know that volume conduction is clearly a significant problem with EEG [13] and also that coherence is a connectivity measure that could be greatly influenced by volume conduction [14, 15]. Furthermore, due to the smaller number of EEG electrodes, they were not able to localize the brain sources which could justify their choice of electrodes of interest. They also did not determine the direction of functional connectivity.

In the current study, we aimed to replicate the results of Mima et al. [12] using high density EEG while participants performed the midline object recognition task and to resolve the limitations of their study. Precisely, we intended to replicate the task-related alpha desynchronisation and the transient increase in interhemispheric coherence during midline object recognition. We further sought to localize the sources in the brain that are crucial for midline object recognition and also to determine the direction of this interhemispheric connectivity.

## Materials and methods

### Participants

Eleven healthy adult volunteers (mean age ± SD = 49.1y ± 14.7y; 6 women) participated in the study. The sample size for the current study was determined by statistical power analysis. Based on the results of Mima et al. [12], we intended to achieve a minimum effect size of 1.25 in event-related coherence, statistical power of 0.80 and threshold alpha of 0.05, which yielded a sample size of 11 estimated based on paired sample t test. Participants were recruited from the healthy volunteer database of the Human Motor Control Section. All participants were right-handed as assessed by Edinburgh Handedness inventory [16]. The protocol was approved by the Combined Neuro Sciences Institutional Review Board (CNS IRB) of the National Institutes Health (NIH) and conformed to the guidelines of the Declaration of Helsinki. All participants gave written informed consent prior to the study. They underwent a physical and neurological examination by a physician and were certified eligible to participate

as a healthy volunteer in the study. They did not have any significant medical/surgical history and did not take any medication that affected the central nervous system.

## Procedure

The participants were seated comfortably on a chair and asked to relax. A 64-channel EEG cap (Braincap, Brain Products GmbH, Germany) was positioned on the participant's head and the skin beneath the electrodes was prepared with abrasive and conductive gel to achieve an impedance less than 5 kohms. EEG was recorded (Brainamp MR, Brain Products GmbH, Germany) throughout the duration of the experiment at a sampling rate of 5000Hz with FCz as reference and AFz as ground electrodes (See Fig 1A for electrode layout).

The experiment consisted of a single session split into 2 blocks, each consisting of 50 trials. In every trial, a visual stimulus, which was an image of either a familiar/meaningful object or an unfamiliar/meaningless object (taken from the standard set of meaningless objects published in [17]), was briefly presented at the centre of the visual field. The participants had to respond by pressing the left mouse button if they recognized the object and the right mouse button if they did not. Visual masks were presented before and after the stimulus to prevent an after-image. The inter-trial interval was 2s. A total of 50 object and 50 meaningless object stimuli were presented randomly over the 2 blocks of the experiment. The course of the experiment is illustrated in Fig 1B.

T1-weighted anatomical magnetic resonance images (MRI) were obtained for all participants from a 3T scanner. Further, the positions of all the EEG electrodes in each participant were captured using an MRI-guided optical neuronavigation system (Brainsight, Rogue Resolutions, Cardiff, UK). Both the individual MRI images and the electrode positions were used for precise source localization.

## Sensor-space analyses

The EEG data analysis was performed using the MATLAB-based toolboxes–Brainstorm [18] and FieldTrip [19], and custom-written scripts in MATLAB. Continuous raw EEG data were bandpass filtered with lower cut-off frequency of 2 Hz, higher cut-off frequency of 50Hz and a stopband attenuation of 60dB. The data was then downsampled to 1000Hz and re-referenced to common average. The data from channels A1 and A2 were removed for all participants. For one of the participant's data from one block, the signal from channel CPz was intermittently saturated and had large fluctuations. Hence, CPz channel was removed from this particular dataset and later interpolated using EEGLAB toolbox. Eye blinks and eye movement artefacts were removed using SSP/ICA methods. An average of 2 ICA components was removed from both datasets of the participants. The data was epoched from -2 seconds to 4 seconds from the time of stimulus onset. The trials which had excessive muscle artefacts were removed by careful visual inspection. The trials from both the blocks were then pooled together and those with correct responses were categorized into objects and meaningless objects datasets. An average of $44 \pm 4$ (mean ± SD; range: 36–49) meaningless objects trials and $44 \pm 3$ (mean ± SD; range: 37–49) object trials per participant were finally used for the analyses.

Time frequency analysis was performed using short-time Fourier transforms with Hanning window tapering for the frequency range of (2–50) Hz with a stepping frequency of 1 Hz. Sliding time windows of 500 ms length were used with 50% overlap of segments, which yielded a frequency resolution of 4 Hz. The resulting values were then log-transformed and normalised to the baseline -500–0 ms to obtain the task-related power change over time. Further, we calculated the alpha band power over the electrodes of interest for five (1 baseline + 4 task-related) 500 ms time windows. We defined such distinct time windows based on the results obtained

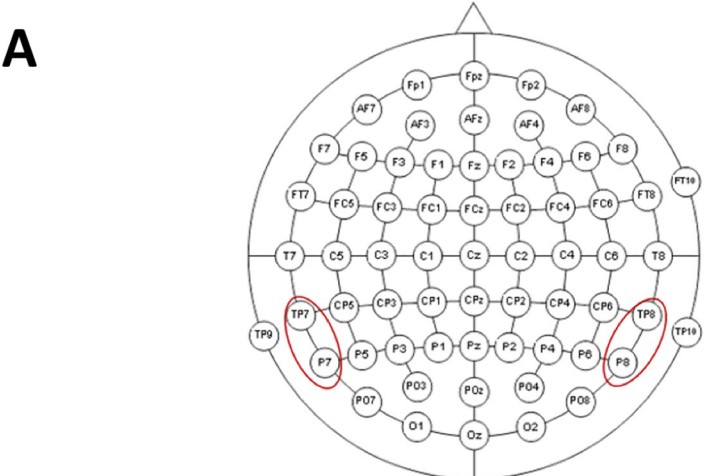

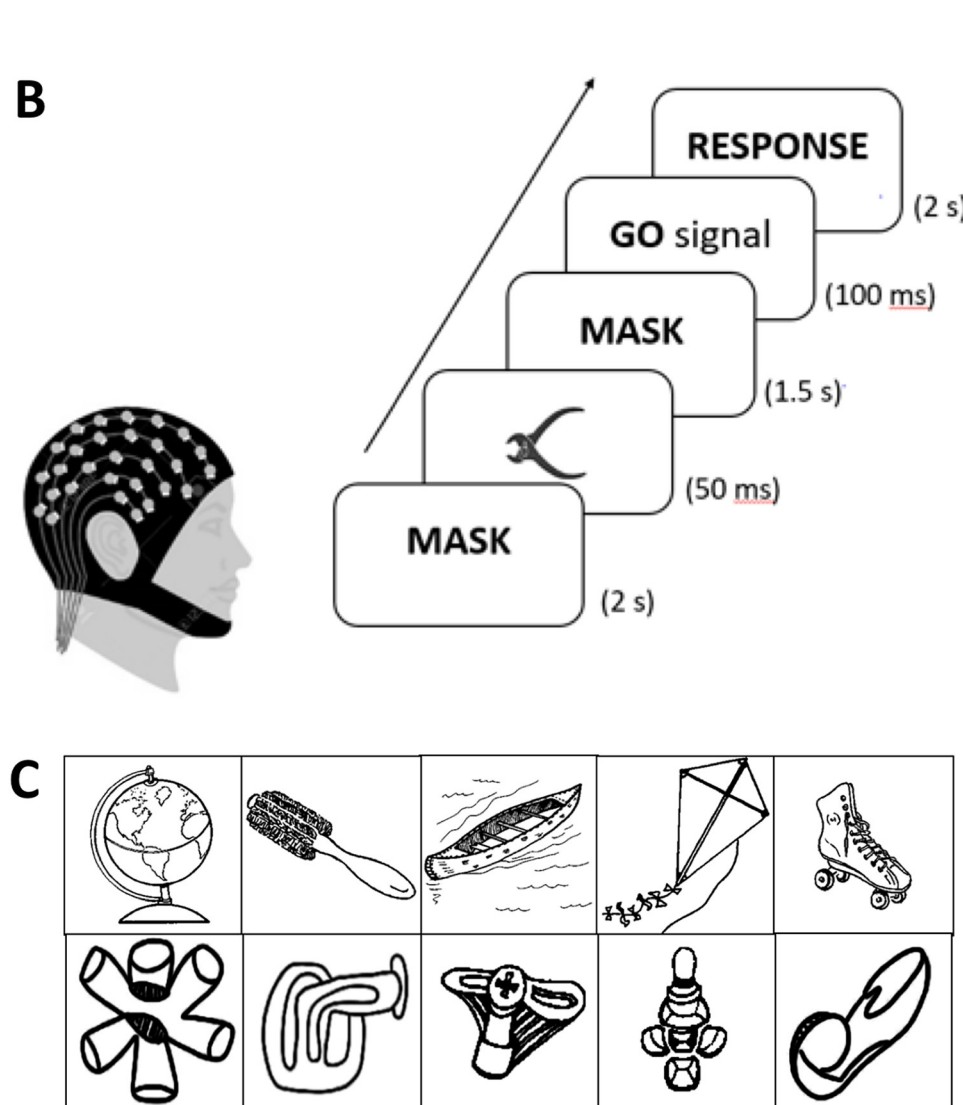

**Fig 1.** (A) Electrode layout used for the experiment. Red ovals indicate the electrodes of interest: TP7, TP8, P7, P8 used in Mima et al. 2001. (B) Illustration of experimental design. (C) Example images of object (top panel) and meaningless object (bottom panel) stimuli.

by Mima and colleagues because we intended to verify if their results were temporally replicable or not.

We calculated interhemispheric alpha band (8–12 Hz) coherence for the electrode pairs of interest (TP7-TP8; P7-P8; TP7-P8 and P7-TP8). These electrodes were chosen by Mima and colleagues based on previous neuroimaging studies which had shown that the regions underlying these electrodes were involved in object recognition. Magnitude-squared coherence [20] is a statistical measure which computes the relation between two signals, like x(t) and y(t), in the frequency domain. We calculated magnitude-squared coherence (Cxy) via the algorithm embedded in Brainstorm toolbox [18], which used the following formula–

$$C_{xy}\,(f) = \frac{|S_{xy}(f)|^2}{S_{xx}(f)S_{yy}(f)}$$

The coherence function used the Fourier transform (f) to compute the spectral densities.

$S_{xy}\,(f)$: - Cross-spectral density

$S_{xx}\,(f)$ and $S_{yy}\,(f)$: - Auto-spectral density

For every participant, the mean alpha (8–12 Hz) coherence for the electrode pairs of interest were calculated separately for five different 500 ms time windows [Baseline = average of -750 to -250 ms & -500 to 0 ms; t1 = 0–500 ms; t2 = 250–750 ms; t3 = 500–1000 ms; t4 = 750–1250 ms]. We then applied inverse hyperbolic tangent transformation to stabilize the variance and calculated task-related coherence (Coh task–Coh baseline) [21].

Additionally, to alleviate the problem of volume conduction, we also calculated imaginary part of coherence [22], which is a more conservative and robust method of evaluating functional connectivity. Imaginary part of coherency, $IC_{xy}$, was calculated as:

$$IC_{xy}\,(f) = \frac{Im\,(S_{xy}\,(f))}{\sqrt{Sxx(f)\,Syy(f)}}$$

Imaginary part of coherency was calculated for the same 5 time windows and the frequency bands, as used for the calculation of magnitude-squared coherence. Using the imaginary part of coherency reduces the possible confounding effects of zero-lag interactions. However, when estimated in sensor-space, even this measure does not provide any meaningful interpretations in terms of underlying cortical sources, but it provides only an estimate of statistical dependencies between sensors [14, 23].

## Source localisation and analysis of source data

Each participant's individual MRI image was used for constructing the forward model and localising the sources of alpha band activity. The fiducial points were set according to the CTF head coordinate system, by computing automatic MNI normalisation [24] followed by manual adjustments. MRI segmentation for each participant was done using the SPM12 algorithm embedded in the Brainstorm interface. Next, the positions of the EEG electrodes which were captured using the neuronavigation system, were used to co-register the MRI anatomy for every participant. The electrodes positions were then manually adjusted to ensure that the sensors were optimally positioned on the scalp surface.

For performing source localisation, the forward model for every participant was constructed using the Symmetric Boundary Element Method of the OpenMEEG [25, 26] software. A head surface was also estimated. A noise-covariance matrix was estimated from the individual participant's EEG recordings, for baseline time window -2 to -0.001 s. Finally, with the help of a linear kernel, the minimum norm estimate method was employed to estimate dynamical Statistical Parametric Mapping (dSPM) [27] sources from every trial's sensor data.

Time frequency analysis was performed on the sources of the individual trials under each condition. To compute time-frequency maps, short-time Fourier transform with Hanning window tapering was performed on 100 ms data segments sliding from -0.5s to 1s with 50% overlap. The obtained maps were then normalised to baseline (-0.5s to 0s) using z-score normalisation.

For determining the interhemispheric coherence, two regions of interest (ROIs) were demarcated based on the results of power spectral analysis. These ROIs were then projected on each participant's individual anatomical cortex surface and the mean time series data from them were extracted for the time range -0.5s to +1s.

Magnitude-squared coherence and imaginary part of coherency values were computed between these two ROIs, participant-wise and for each condition. Spectral Granger causality [28, 29] values were computed in the frequency domain for both left ROI-to-right ROI and right ROI-to-left ROI directions, using FieldTrip toolbox [19].

## Statistical analysis

**Sensor space.** To replicate the results of Mima and colleagues, we extracted the task-related alpha power and coherence values, and performed statistical analysis using IBM SPSS Statistics version 23. For alpha power, repeated measures ANOVA was applied with mean task-related power over electrodes of interest as dependent variable, and stimulus (object/ meaningless object) and time (t1/ t2/ t3/ t4) as independent variables. For coherence data, mean task-related alpha band coherence over the electrode pairs of interest was used as the dependent variable. The data were tested for sphericity using Mauchly's test and Greenhouse-Geisser correction was applied wherever needed. A p value of 0.05 was considered to be statistically significant. Further, relevant post hoc pairwise comparisons were made and Bonferroni correction was applied to correct for multiple comparisons.

**Source space.** Using the results from the time-frequency analysis in the source space, a paired permutation test [30] (via algorithm embedded in Brainstorm toolbox) was performed for the time window of -0.5s to 1s. The threshold p-value was set at 0.05 and corrected for multiple comparisons. The obtained results were projected on a template anatomy cortex for visualisation.

## Data-driven analysis

Mima et al. used only four electrodes of interest for their analysis. They were selected on the basis of prior neuroimaging studies that showed those locations to be involved in object recognition. Since we could not replicate their coherence results, we wanted to verify whether this selection of electrodes was appropriate using a data-driven approach. We applied non-parametric cluster based permutation tests [31] on the alpha power data from all 64 channels and across all time points, and looked for time points and electrodes that showed significant differences in the task-related alpha power change for object versus meaningless object stimuli. A similar data-driven statistical approach [32] was also employed to identify electrode pairs of interest that showed significant differences in coherence.

## Results

All participants tolerated the procedure well and no adverse events occurred. An average of 89 ± 6 (mean ± SD, range: 78–98) trials (including object and meaningless object trials) per participant were used for final analysis.

### Sensor space

We did not observe any significant difference in the baseline alpha power ($t(10) = -1.50964$; $p = 0.162065$) and coherence ($t(10) = -0.75202$; $p = 0.469367$) for object versus meaningless object stimuli. Time frequency plots for objects and meaningless objects over the electrodes of interest (TP7, TP8, P7, P8) showed a clear decrease in task-related power for both object and meaningless object stimuli in the alpha (8-13Hz) frequency band (see Fig 2). We were able to replicate the non-specific alpha desynchronisation, as reported by Mima et al. [12]. ANOVA showed significant main effect of time ($F(1.881,18.814) = 16.409$, $p<0.001$), but not stimulus ($F(1,10) = 0.631$, $p = 0.445$). We did not find a significant time x stimulus interaction ($F(4,40) = 0.901$, $p = 0.472$). Post-hoc pairwise comparisons using paired t-test showed that for both object and meaningless object stimuli, alpha power was significantly lower (objects: $t(10) = 5.401$; $p<0.001$ meaningless objects: $t(10) = 3.647$; $p<0.005$) during t2 (250–750 ms) alone (Fig 3).

With the connectivity analysis, however, we could not reproduce the increase in interhemispheric magnitude-squared coherence that was reported by Mima and colleagues. ANOVA showed neither time ($F(2.596,25.958) = 0.664$, $p = 0.561$) nor stimulus ($F(1,10) = 0.385$, $p = 0.549$) nor their interaction ($F(2.708,27.078) = 2.024$, $p = 0.168$) to be significant (Fig 4A). For imaginary part of coherency (Fig 4B), ANOVA showed no significant main effects but significant stimulus x time interaction ($F(4,40) = 2.928$, $p = 0.033$). Further, post-hoc pairwise comparisons with paired t-test showed a near-significant difference between objects and meaningless objects conditions in 500–1000 ms time window ($t(10) = 2.183$; $p = 0.054$).

A non-parametric cluster-based permutation test showed significant alpha desynchronisation over C3, P3, Pz and O1 electrodes for time points 600 and 700ms ($p = 0.02$) (Fig 5A and 5B). However, similar analysis for coherence data did not reveal any significant electrode clusters.

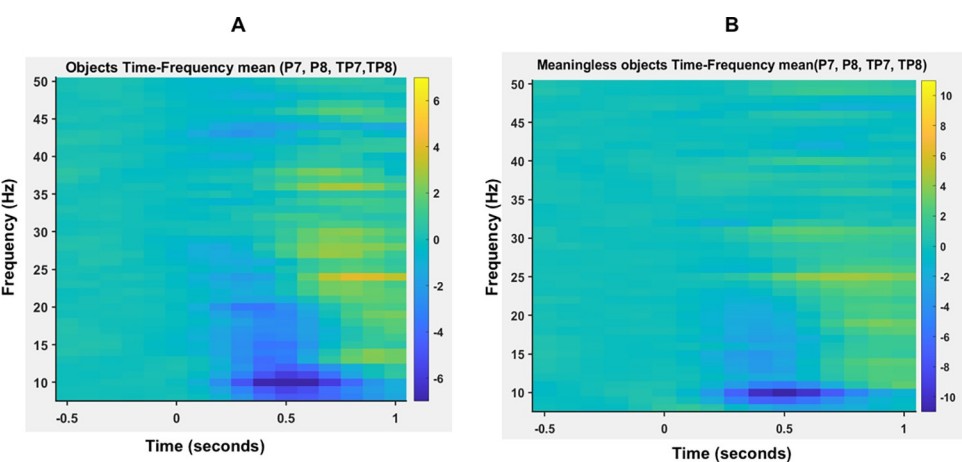

**Fig 2. Frequency power spectrum averaged over four electrodes of interests (TP7, TP8, P7, P8) for object (2A) and meaningless object (2B) stimuli plotted across time.**

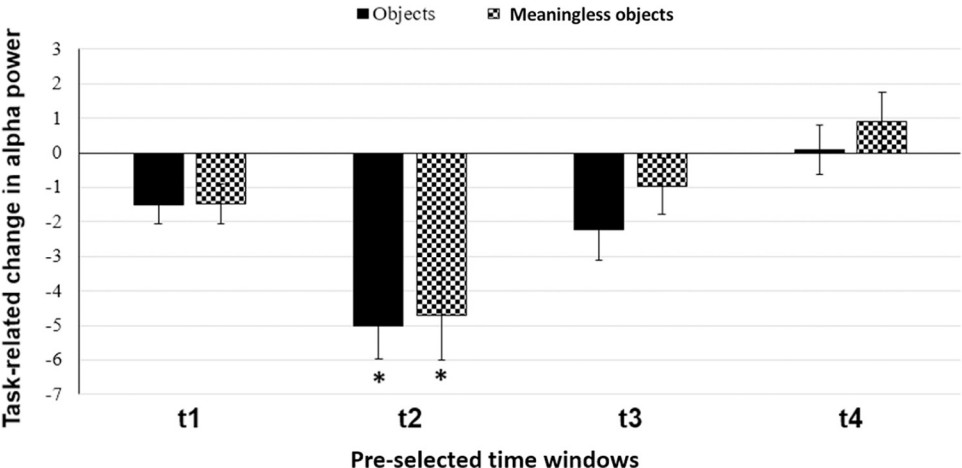

**Fig 3. Task-related change in alpha band power averaged over four electrodes of interest (TP7, TP8, P7, P8) and plotted for different time windows.** Solid black bars represent object stimuli; checked bars represent meaningless object stimuli. t1 = 0–500 ms; t2 = 250–750 ms; t3 = 500–1000 ms; t4 = 750–1250 ms. Error bars are SEM. Asterisks indicate p<0.05.

## Source space

Paired permutation test was performed on the power spectrum data (averaged across time and frequency) from the source grids for the alpha band (8–12 Hz) and for the 500–750 ms time window. The results from the comparison of 'objects' vs 'meaningless objects' were projected on a template cortex (Fig 6), for visualisation of the regions which showed significant difference between conditions. As shown in the figure, the left occipito-temporal region showed significant desynchronisation during object recognition. Based on this result, the ROIs were demarcated such that the first ROI covered the left occipito-temporal region and the second ROI covered the homologous region in the right hemisphere. Fig 7 shows the two ROIs comprising equal number of vertices, between which connectivity analysis was performed. To evaluate the connectivity between the defined ROIs, magnitude-squared coherence was computed in the alpha band for five different time windows. ANOVA did not show a significant main effect of stimulus (F(1, 10) = 1.813, p = 0.208) or stimulus x time interaction (F(4, 40) = 0.136, p = 0.968). However, main effect of time (F(2.384, 23.842) = 8.592, p = 0.001) was significant. Post hoc analysis revealed the increase in coherence with respect to the baseline, during the time window of 0–250 ms to be statistically significant (objects: t(10) = -2.28554; p = 0.045 meaningless objects: t(10) = -3.55515; p = 0.005) (Fig 8A).

In order to minimize volume conduction effects that may give rise to spurious connectivity between regions, imaginary part of coherency was computed [22]. The results of ANOVA revealed a significant main effect of stimulus (F(1,10) = 6.128, p = 0.033) alone. Main effect of time (F(4,40) = 0.448, p = 0.773) and stimulus x time interaction (F(4,40) = 1.193, p = 0.329) were not significant. Post-hoc pairwise comparisons showed that there was a significant increase in imaginary part of coherency for object stimuli in the 250–500 ms time window (t (10) = 2.219; p = 0.05) (Fig 8B).

Further, we determined the direction of this transient increase in interhemispheric connectivity. Granger causality values were computed between the ROIs, in both left-to-right and right-to-left directions for the time window of 250–500 ms (Fig 9) as the imaginary part of coherency was significantly different between conditions only during this time period. ANOVA results showed significant main effect of direction (F(1,10) = 7.386, p = 0.022) but

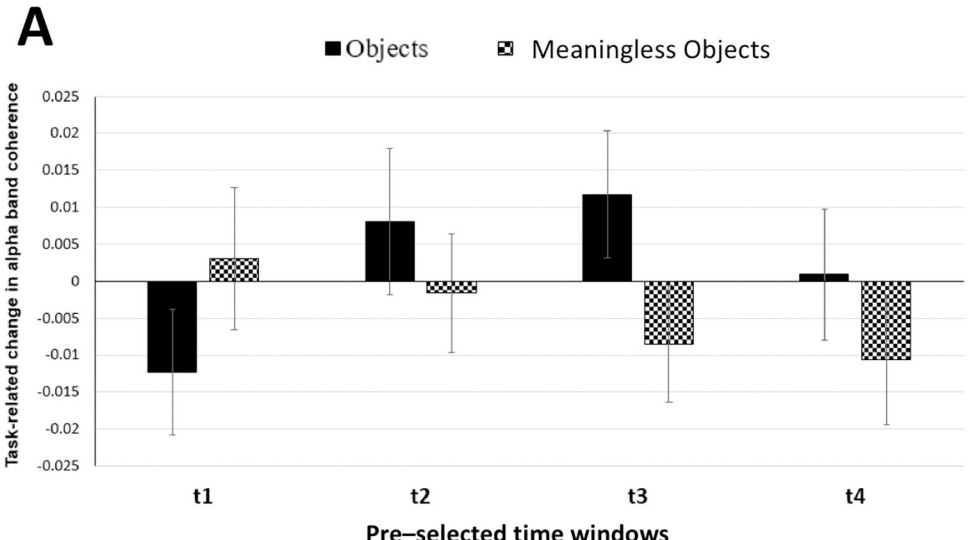

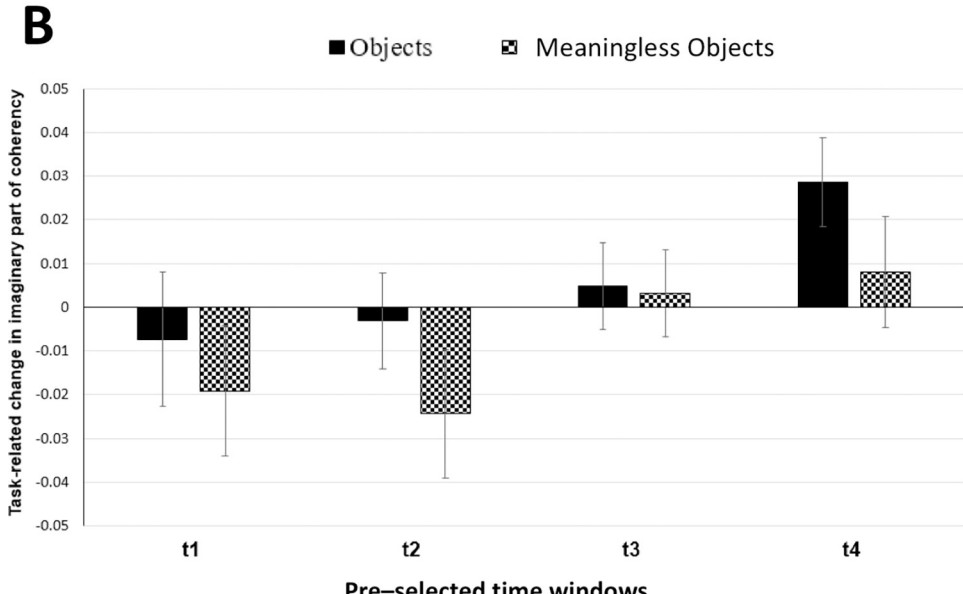

**Fig 4.** (A) Task-related change in alpha band coherence averaged for four electrode pairs of interest (TP7-TP8, TP7-P8, P7-TP8 and P7-P8) pairs and plotted for four different time windows. Solid black bars represent object stimuli; checked bars represent meaningless object stimuli. t1 = 0–500 ms; t2 = 250–750 ms; t3 = 500–1000 ms; t4 = 750–1250 ms. Error bars are SEM. (B) Task-related change in imaginary part of coherency for alpha band averaged for four electrode pairs of interest (TP7-TP8, TP7-P8, P7-TP8 and P7-P8) and plotted for four different time windows. Solid black bars represent object stimuli; checked bars represent meaningless object stimuli. t1 = 0–500 ms; t2 = 250–750 ms; t3 = 500–1000 ms; t4 = 750–1250 ms. Error bars are SEM.

not stimulus (F(1, 10) = 0.068, p = 0.799) or stimulus x direction interaction (F(1,10) = 1.374, p = 0.268). Although left-to-right causality was observed to be more than right-to-left causality for both object and meaningless object stimuli, post-hoc pairwise comparisons using paired t test showed that left-to-right directional connectivity was significantly higher only for object stimuli (t(10) = 2.605, p = 0.026) but not for meaningless objects (t(10) = 1.197, p = 0.259).

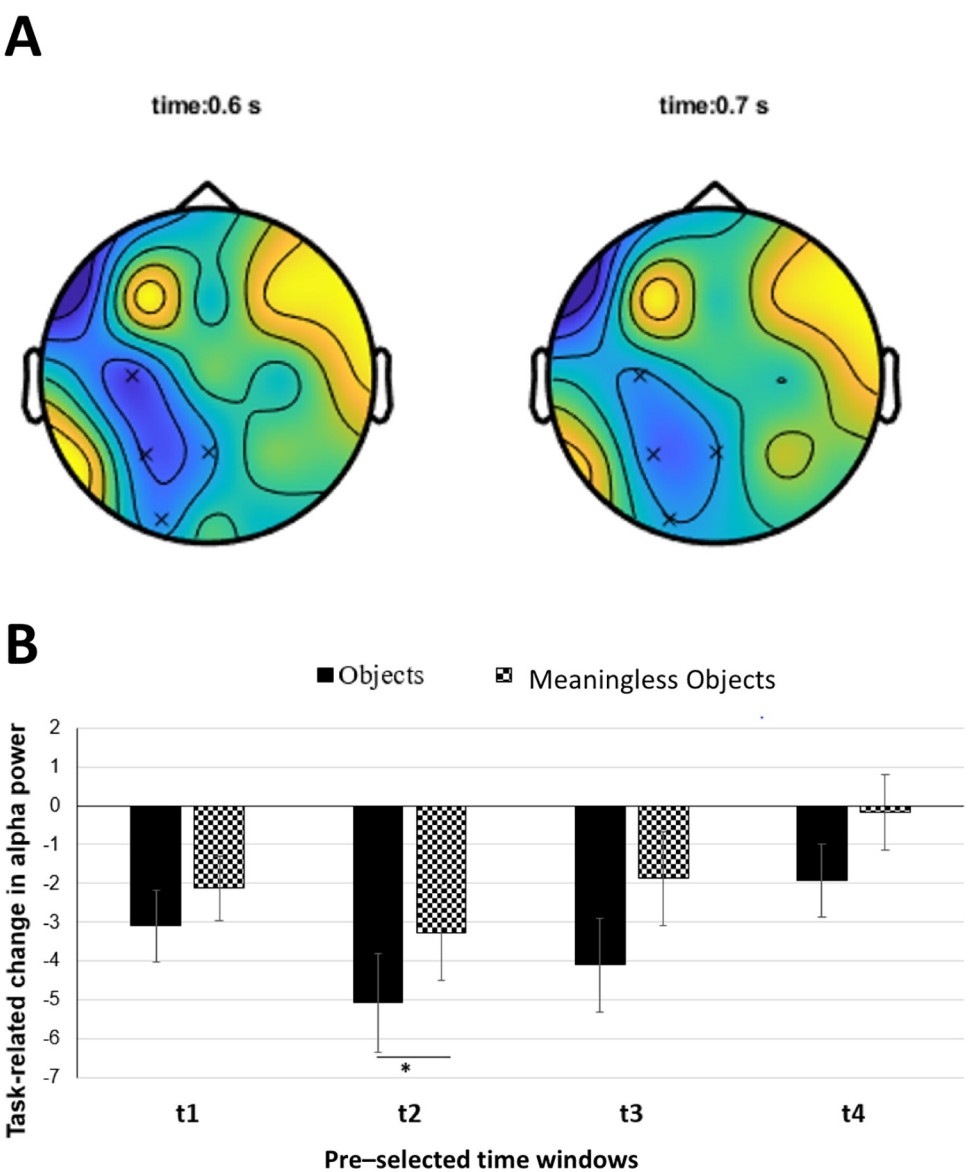

**Fig 5.** (A) Results of non-parametric cluster-based permutation analysis of power spectrum data for alpha band. Asterisks indicate electrodes that showed significant difference (p<0.05) in alpha power between the two stimulus conditions (Objects minus meaningless objects) that occurred at 0.6s (left) and 0.7s (right). The prominent cluster visible over the left occipito-temporal region included C3, P3, Pz and O1 electrodes. (B) Task-related change in alpha power averaged for the significant electrode cluster (C3, P3, Pz and O1) and plotted for four time windows. Solid black bars represent object stimuli; checked bars represent meaningless object stimuli. t1 = 0–500 ms; t2 = 250–750 ms; t3 = $C_{xy}$ (f) = 500–1000 ms; t4 = 750–1250 ms. Error bars are SEM. Asterisks indicate p<0.05.

## Discussion

In the current study, we were able to partially replicate the results of Mima et al. [12]. We confirmed that the stimulus non-specific alpha desynchronisation occurs in the time window around 500ms. However, we could not replicate the transient increase in interhemispheric alpha coherence associated with object recognition using the same electrode pairs of interest. Further, with detailed analysis using a data-driven approach we showed that there is significantly high alpha desynchronisation associated with object recognition over C3, P3, Pz and O1

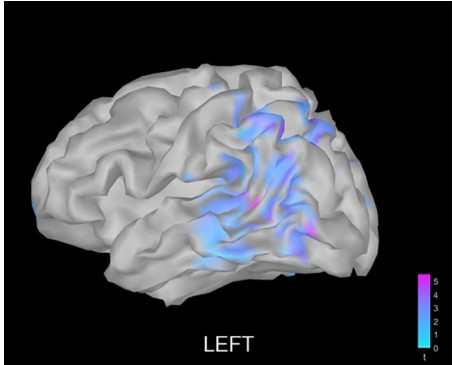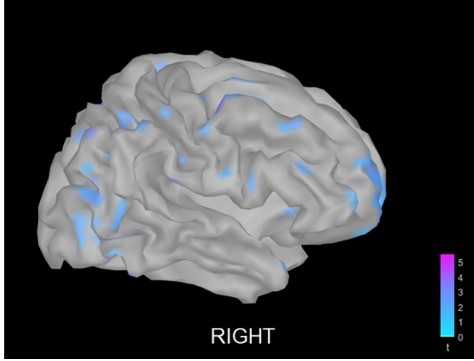

**Fig 6. Results of paired permutation test (objects vs meaningless objects) performed on alpha power spectrum of sources and projected on a template anatomy cortex.**

electrode cluster. We localised the source of this alpha desynchronisation to be in the left occipito-temporal region and showed that interhemispheric connectivity (imaginary part of coherency) between the homologous regions increased significantly for object recognition during the 250–500 ms time window. Using Granger causality, we also showed that the interhemispheric connectivity associated with object recognition is driven by the left occipito-temporal region.

The possible reason why we could not replicate the sensor-space coherence results of Mima's study could be because of the high inter-individual variability in the data especially with a sample size of 8 in Mima's study [12]. Using the effect size and variability in their results, we performed a statistical power analysis and accordingly used 11 participants for our study. Notably, we used more appropriate and powerful analytical tools which have demonstrated convincingly the transient increase in interhemispheric connectivity during midline object recognition.

As in our study, recording with a higher-density EEG system is known to enable better capturing of brain activity in the sensor space itself as we can record signals from more distributed regions of the brain [33]. Also, our efforts in using the same protocol of data analysis as Mima's study, not only validated their results but also gave more definition and accuracy to their observations. Very relevant in this context would be a study by Robinson et al. [34],

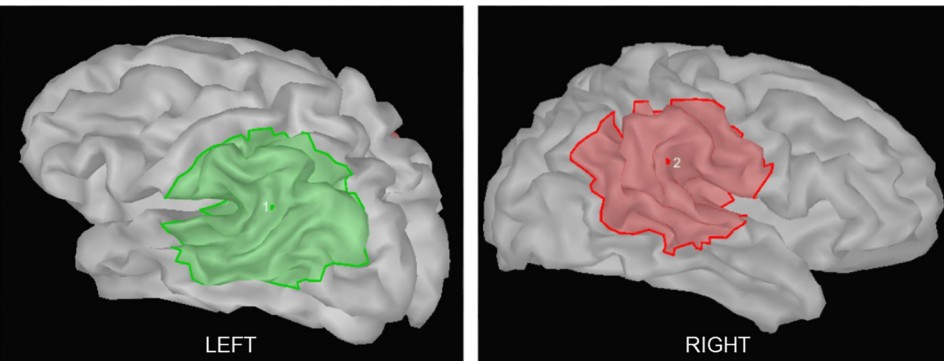

**Fig 7. Regions of interest (ROIs) demarcated based on the results of time-frequency analysis for source space.** The left ROI encompassed the occipito-temporal region in the left hemisphere that showed significant alpha desynchronisation associated with object recognition and the right ROI included the homologous region in the right hemisphere.

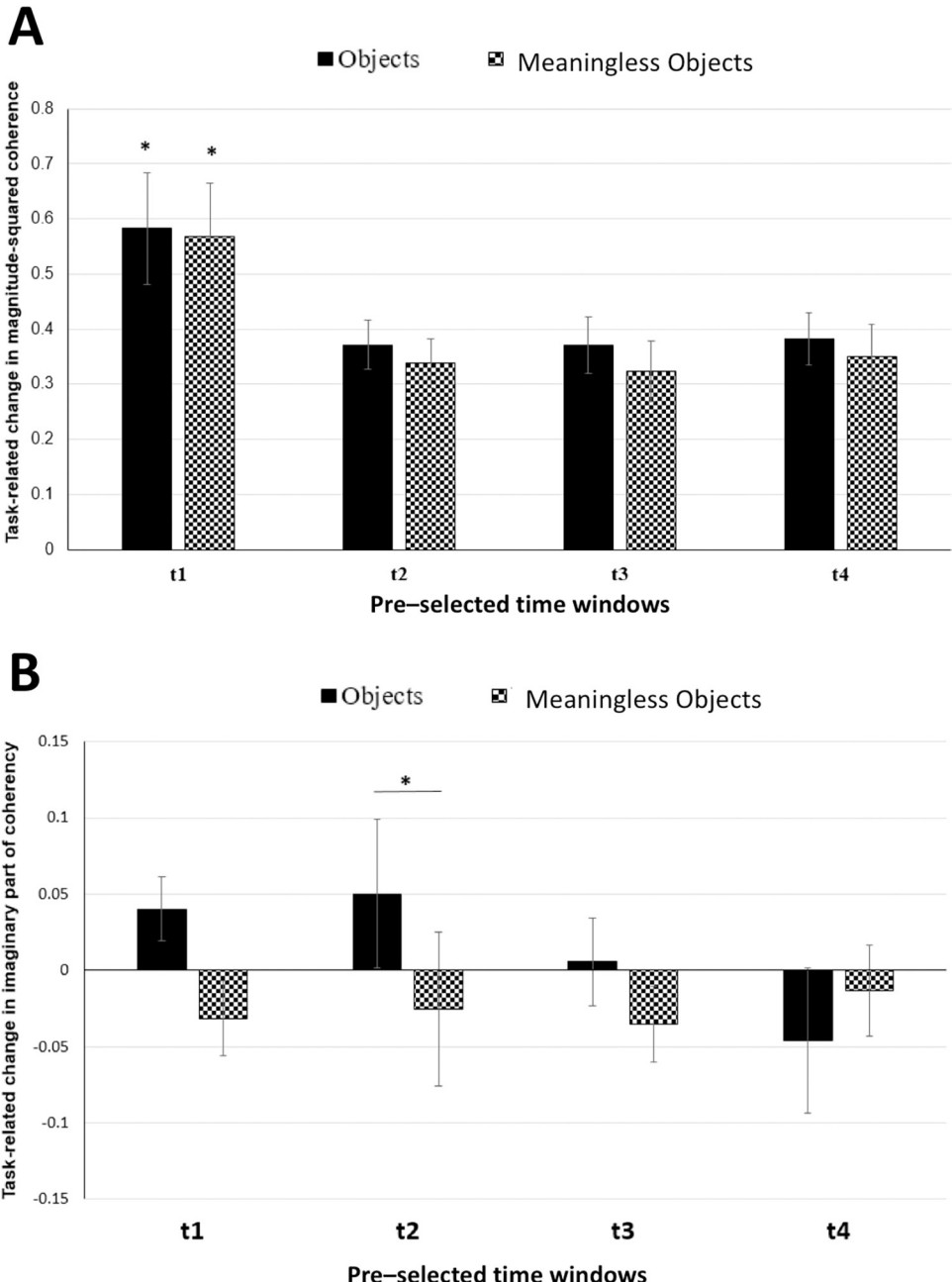

**Fig 8.** (A) Task-related change in alpha band coherence between the left ROI and right ROI in source space and plotted for four different time windows. Solid black bars represent object stimuli; checked bars represent meaningless object stimuli. t1 = 0–250 ms; t2 = 250–500 ms; t3 = 500–750 ms; t4 = 750–1000 ms. Error bars are SEM. Asterisks indicate p<0.05. (B) Task-related change in imaginary part of coherency for alpha band between the left ROI and right ROI in source space and plotted for four different time windows. Solid black bars represent object stimuli; checked bars represent meaningless object stimuli. T1 = 0–250 ms; t2 = 250–500 ms; t3 = 500–750 ms; t4 = 750–1000 ms. Error bars are SEM. Asterisks indicate p<0.05.

which showed both experimentally and computationally that 128-channel EEG captured considerably more neural information from the visual cortex than 33-channel configurations designed from 128-channel cap. Therefore, our study using 64-channel EEG allowed for a more precise and accurate localisation of the sources of activity, compared to 29-channel EEG

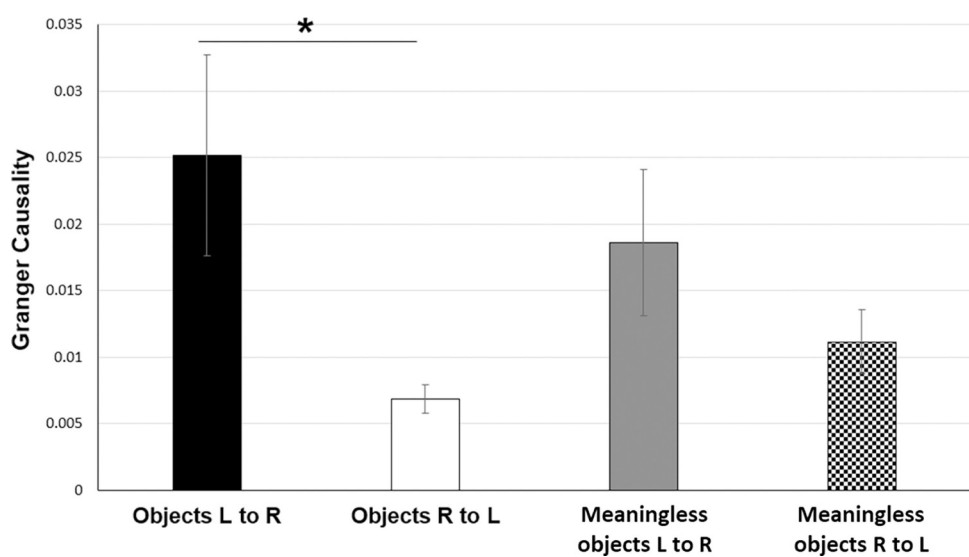

**Fig 9. Results of Granger causality (GC) analysis between left and right ROI in source space for the time window of 250–500 ms.** Solid black bar represents Left to Right GC for object stimuli; solid white bar represents Right to Left GC for object stimuli; solid grey bar represents Left to Right GC for meaningless object stimuli; checked bar represents Right to Left GC for meaningless object stimuli. Error bars are SEM. Asterisks indicate $p < 0.05$.

used by Mima et al. The advantage of using an EEG system with a higher density of electrodes has been acknowledged by many researchers in the past. A study by Lantz et al [35] had measured EEG under clinical conditions and found that increasing the number of electrodes from 31 to 63 to 128, considerably increased the source localisation accuracy. Another valuable component in the current study is the time-frequency and connectivity analyses performed using a data-driven approach [36], so as to get a global insight into the brain activity during object recognition and also to confirm that the electrodes chosen by Mima's study were appropriate. We clearly show in our results that by choosing a predetermined set of electrodes, Mima et al. had overlooked the stimulus-specific alpha desynchronisation in a neighbouring electrode cluster which we have identified in the current study. This also reiterates the inherent problem of volume conduction with EEG. Additionally, in our study, we support the sensor-space results with source localisation via estimation of head model from individual participant's MRI. The advantage of EEG source localisation over other neuroimaging modalities, lies not only in its reasonably precise localisation of sources, but also its high temporal resolution, which enables us to distinctly track the temporal propagation of information from the origin of activity to other regions in distributed brain networks [37, 38].

At the time when Mima et al. [12] published their results, coherence (cross-spectra normalised by auto-spectra), was a widely used measure for estimating connectivity between brain regions at the sensor level. However, this method has a major drawback that it does not tackle the problem of volume conduction [14] which is the superposition of many sources on a single overlying sensor. A few years later in 2004, a study by Nolte et al. [22], introduced the use of the 'imaginary part of coherency' measure which eliminates all sources of extraneous activity that are a consequence of instantaneous activity and captures true source interactions with a time lag. This method of estimating connectivity, although very conservative, is much more robust [39] than coherence as it has a 100% positive predictive value, i.e., whenever it produces a significant value, it indicates presence of coordinated activity between distinct sources. It does not capture self-correlation or coordination of the same sources [15, 40, 41]. In our study,

we evaluated both magnitude-squared coherence [20, 42] and imaginary part of coherency [22] in the sensor space as well as source space. Although we did not see significant results with magnitude-squared coherence, evaluating imaginary part of coherency yielded significant results in the time window of our interest. Having used a more conservative measure to bypass one of the biggest drawbacks of EEG, we can confidently say that our results are more representative of true and robust functional connectivity [22, 39] since it excludes self-correlations arising from volume conduction. One of the reasons why we could not obtain significant magnitude-squared coherence could be due to considerable volume conduction. Since we were evaluating connectivity in the alpha frequency range, there is a possibility that a common alpha rhythm, probably from the occipital cortex, was picked up by all the electrodes. Our results showing significant increase in imaginary part of coherency within the first 500 ms of stimulus presentation, confirms that there is definitely an enhanced flow of information between the two hemispheres during that time window. This is also in accordance with the results of Mima et al. which also identified that the coherence change preceded the power change. We have not fully exploited the advantage of high temporal resolution of the technique. However, we have been able to achieve precise localization of activity and narrow down our findings to a temporal resolution of 250ms. We had to pool our data to 250ms time windows to decrease the number of multiple comparisons. Future studies can focus on further improving upon the current methodology by using MEG to decrease volume conduction effects or increasing the temporal resolution of the effects.

The ROI that we identified to be specific for object recognition lies in the occipito-temporal region, falling in the ventral (what) pathway of visual processing [43]. The visual information from the retina reaches the striate cortex or V1 after a few synapses. Further processing of this information occurs via two main pathways—either a dorsal (occipito-parietal stream) path from V1 to posterior regions of the parietal lobe or a ventral (occipito-temporal stream) path from V1 to the temporal lobe of the brain. It is known that the ventral stream is associated with object recognition; i.e., for determining 'what' the object is, whereas the dorsal stream is crucial for spatial perception–for determining 'where' an object is. The region which showed significant alpha desynchronisation during midline object recognition in our study clearly falls in the left hemisphere's ventral stream of visual processing [1, 43]. This confirms that our region of interest is surely relevant to object recognition. However, our results from time-frequency analysis in the source space showed that the significant change in alpha power is more pronounced in the left hemisphere than in the right hemisphere. We did not see significant change in alpha power during object recognition over the right hemisphere. Existing literature provides two possible reasons for this observation–ocular dominance and attentional bias. Ocular dominance or eyedness is the tendency to prefer visual input from one eye than the other eye [44]. Studies have shown how handedness is related to one's dominant eye; i.e., right-handed people tend to have their right eye as the dominant one [45, 46]. As all the participants in our study were right-handed, it is possible that they had right-eye dominance and therefore a perceptual bias in the left hemisphere [47]. Secondly, both modelling [48] and empirical studies [49] in the past have shown evidence for a hemispheric bias (theory of *interhemispheric competition)* in attentional mechanisms and it is known that the rightward bias of the left hemisphere is stronger than the leftward bias of the right hemisphere. This asymmetry leads to a left-hemispheric dominance in attention. Alpha desynchronisation is representative of an active state with respect to visual processing [50] as decrease in alpha power is associated with increased attention [51]. Thus, significant decrease of alpha power over only the left hemisphere may suggest that attentional networks were active during the time frame of recognition [52]. Finally, recognition of a meaningful object involves matching of the visual information with prior semantic knowledge of the object's features and retrieving the object's name

correctly which is a language skill. As previous studies have shown that semantic processing and language lateralization are predominantly localised in the left hemisphere [53–55], it seems plausible that we clearly observed dominance of left hemispheric activity during object recognition [56].

In conclusion, our results confirm that midline object recognition requires integration of information from the two hemispheres. This interhemispheric exchange of information is mediated by transient synchronisation of neuronal activity between the ventral pathways of visual processing. Interhemispheric coherence has relevance not only in the visual sensory system but is also crucial for a broad repertoire of cognitive functions including language, visuospatial attention and manual preference [57–59]. Consequently, disruption of interhemispheric coherence may impact behavioural [60, 61] and cognitive functioning. Thus, our study may have implications in many neuropsychiatric disorders like schizophrenia [62–65], stroke [66, 67], focal hand dystonia [68] where interhemispheric coherence is known to be impaired. Future studies may attempt to boost the impaired interhemispheric connectivity, using non-invasive brain stimulation methods [69, 70] like rhythmic transcranial magnetic stimulation or transcranial alternating current stimulation and evaluate the impact on behaviour.

## Author Contributions

**Conceptualization:** Silvina G. Horovitz, Mark Hallett, Nivethida Thirugnanasambandam.

**Data curation:** Anwesha Das, Alexandra Mandel, Nivethida Thirugnanasambandam.

**Formal analysis:** Anwesha Das, Nivethida Thirugnanasambandam.

**Funding acquisition:** Mark Hallett.

**Investigation:** Hitoshi Shitara, Traian Popa, Mark Hallett, Nivethida Thirugnanasambandam.

**Methodology:** Hitoshi Shitara, Traian Popa, Silvina G. Horovitz, Mark Hallett, Nivethida Thirugnanasambandam.

**Project administration:** Nivethida Thirugnanasambandam.

**Supervision:** Silvina G. Horovitz, Mark Hallett, Nivethida Thirugnanasambandam.

**Validation:** Nivethida Thirugnanasambandam.

**Visualization:** Nivethida Thirugnanasambandam.

**Writing – original draft:** Anwesha Das, Nivethida Thirugnanasambandam.

**Writing – review & editing:** Anwesha Das, Alexandra Mandel, Hitoshi Shitara, Traian Popa, Silvina G. Horovitz, Mark Hallett, Nivethida Thirugnanasambandam.

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
