## [Decision Letter · Decision Letter 0]

31 Jan 2022

PONE-D-21-26846Evaluating interhemispheric connectivity during midline object recognition using EEGPLOS ONE

Dear Dr. Thirugnanasambandam,

Thank you for submitting your manuscript to PLOS ONE. After careful consideration, we feel that it has merit but does not fully meet PLOS ONE’s publication criteria as it currently stands. Therefore, we invite you to submit a revised version of the manuscript that addresses the points raised during the review process.

We look forward to receiving your revised manuscript.

Kind regards,

Kiyoshi Nakahara, PhD

Academic Editor

PLOS ONE

https://journals.plos.org/plosone/s/file?id=ba62/PLOSOne_formatting_sample_title_authors_affiliations.pdf”

2. Please change "female” or "male" to "woman” or "man" as appropriate, when used as a noun (see for instance https://apastyle.apa.org/style-grammar-guidelines/bias-free-language/gender).

“The work was supported by NINDS intramural research program. NT was funded by Ruth L. Kirschtein NRSA award from NINDS (2013-2016) and DBT/WT India Alliance fellowship grant IA/CPHI/16/1/502624 (2017-present). AD was funded by the Master’s research fellowship from DBT.”

“The work was supported by NINDS intramural research program. NT was funded by Ruth L. Kirschtein NRSA award from NINDS (2013-2016) and DBT/WT India Alliance fellowship grant IA/CPHI/16/1/502624 (2017-present). AD was funded by the Master’s research fellowship from DBT.

Reviewers' comments:

Reviewer's Responses to Questions

**Comments to the Author**

1. Is the manuscript technically sound, and do the data support the conclusions?

Reviewer #1: Partly

2. Has the statistical analysis been performed appropriately and rigorously? 

Reviewer #1: Yes

3. Have the authors made all data underlying the findings in their manuscript fully available?

Reviewer #1: Yes

4. Is the manuscript presented in an intelligible fashion and written in standard English?

Reviewer #1: Yes

5. Review Comments to the Author

Reviewer #1: Das and colleagues aimed to replicate the results of a previous EEG study (Mima et al., J. Neurosci., 2001), which showed that midline recognition of familiar, visual objects induces a transient interhemispheric coherence in the alpha-band; while such interhemispheric synchronization was absent during the presentation of meaningless objects or passive viewing. In the present study, Das et al. assessed interhemispheric coherence during midline object recognition using EEG, and they evaluated also the interhemispheric functional connectivity, after EEG source reconstruction. The Authors replicated the alpha desynchronization effect from the 2001 study, but not the interhemispheric coherence results. Nonetheless, they found modulations in the imaginary part of coherency and Granger causality estimates between left and right occipito-temporal regions.

I agree that the use of more advanced methods and connectivity measures is interesting to investigate object recognition and it may help to provide novel insights into this neural mechanism, and the Authors should be applauded for pursuing this path. However, I have some major concerns that make the current work not suitable for publication in its current form. My concerns are listed below in order of occurrence. I hope that these points will help the Authors to improve their work and manuscript.

<abstract>

* It is unclear whether the imaginary coherency effects are also specific for the alpha-band. The Authors should explicitly mention that in the Abstract.

<introduction>

* The Authors use interchangeably the words “synchronization” and “synchronisation”. Please be consistent. In general, a final proofreading is needed to correct typos and make the grammar more consistent.

* In the first paragraph, the Authors introduce the term “alpha rhythm” without specifying the frequency range. These details are important, especially for a broader, less expert readership.

Also, the use of parentheses to report time windows is confusing. I suggest to avoid that, and instead report simply the extremes of each time interval using an en dash (e.g., “117–373 ms”).

* Mima et al. employed familiar and unfamiliar/meaningless objects. I am not sure what “non-object stimuli” are. I recommend the Authors to clarify the design of the previous study and report its results using unambiguous terminology. This will also help the reader to understand the rationale and details of the current work.

<materials and="" methods="">

* (minor) Figure 1. More or less 50% of this figure is occupied by the electrodes layout (1A), which does not really add any useful information and it is not central for the description of the procedure. More space should be given to the design of the experiment and the stimuli (1B-C).

* The Authors mentioned that anatomical MRI images “were either MPRAGE or 3D-T1 images from a 3T scanner”. What does that mean? Did the Authors used different sequences for anatomical image acquisition across subjects? Furthermore, several technical details are missing. What parameters were used for data acquisition? What is the model and manufacturer of the scanner?

* (minor) Sensor data analysis. I suggest to change the name of the subsection to “Sensor-space analyses”.

* Sensor data analysis. For one subject in one session, one channel was rejected and interpolated. However, no motivation for doing that has been provided. Was the channel contaminated by excessive artifacts during the recording? If that is the case, how were these identified? Please motivate this choice.

* (minor) Sensor data analysis. The Authors report mean and standard deviation across subjects of the number of trials used for the successive analyses. In addition to the SD, the range (minimum-maximum across subjects) may actually provide a more informative measure of the amount of trials surviving preprocessing.

* Sensor data analysis. The Authors should specify the frequency range for the estimation of alpha-band coherence. This information is currently missing in the Methods section.

* Using the imaginary part of coherency reduces the possible confounding effects of zero-lag interactions. However, when estimated in sensor-space, even this measure does not provide any meaningful interpretations in terms of underlying cortical sources, but it provides only an estimate of statistical dependencies between sensors (see Brunner et al., Front. Comput. Neurosci., 2016; Van de Steen et al., Brain Topogr., 2016). To avoid any confusion, the authors should clarify that.

* Source localization. How was the noise covariance matrix estimated? Which time window have the Authors employed for estimation?

* Source localization. The Authors mentioned that Granger causality estimates were derived to assess the interactions between left and right ROIs. It is unclear what variant of Granger causality was used: time-domain definition, Geweke’s frequency-domain definition, or else? References are also missing. Without these details, it is impossible to assess the quality of the methodological procedures.

* Statistical analysis: Sensor space. It is unclear why the Authors restricted their analysis to the alpha-band. I understand the goal to replicate the previous findings, but thanks to modern computers and cluster-based permutation tests for statistical comparison (e.g., see Maris and Oostenveld, J. Neurosci. Methods, 2007), a full time-frequency analysis in sensor-space is called for.

<results>

* Figure 2. The font of axes and colorbar ticks’ labels should be increased. Units are missing for the labels of the x and y axis. Please use the letters A-B to distinguish between the left and right panel in the figure. I also recommend to center the scale of the colorbar around zero (i.e., no increase/decrease) to improve interpretability of the results.

* Figure 3. In addition to describing the difference in bars’ texture between the two types of stimuli (solid black/checked) in the caption, please add a legend showing such difference. This information should be immediately clear from the figure. Furthermore, I suggest to rename the label of x-axis to “Preselected time windows” or something like that.

* Figure 4. Same comments as for Figure 3 (see above).

* Sensor-space coherence/iCoh results. Something unclear to me is the reason why the Authors decide to report and comment only the statistical comparison with respect to baseline if it is not the best at capturing the main results that they are trying to convey (i.e., differences between familiar and unfamiliar objects). The Authors did such comparison only for the power (see Figure 5). Unless the Authors have a specific reason to avoid that, I suggest adding to the statistical design the comparison familiar vs. unfamiliar/meaningless objects. This analysis could be done in the entire time-frequency space, by using the nonparametric cluster-based permutation approach (as I mentioned in a previous point).

* Sensor-space results. Like for the other analyses, it is unclear why the Authors restricted the comparison in few time windows and one frequency band.

Also, the Authors mention that “In order to get a more conservative and robust estimation of the coherence between the two regions, imaginary part of coherency was computed.” This sentence is misleading and should be rephrased. Coherence and iCoh measure essentially two different things.

* Figure 8. Same comments as for Figure 3 (see above).

* Granger causality results. Without specifying how Granger causality was estimated it is impossible to assess the reliability of the results shown.

<discussion>

* I agree that the advantage of using EEG source reconstruction is its high temporal resolution, combined with a reasonably acceptable localization of cortical sources of activity. However, the Authors did not take full advantage of this characteristics of the EEG signals, by restricting their analyses to four windows of interest.

* What is the definition of “true functional connectivity” for the Authors? I suggest to expand on that. And, in general, I recommend to be a bit more cautious in drawing certain conclusions.</discussion></results></materials></introduction></abstract>

6. PLOS authors have the option to publish the peer review history of their article (what does this mean?). If published, this will include your full peer review and any attached files.

Reviewer #1: No

---

## [Author Response · Author response to Decision Letter 0]

21 Mar 2022

Reviewer #1: Das and colleagues aimed to replicate the results of a previous EEG study (Mima et al., J. Neurosci., 2001), which showed that midline recognition of familiar, visual objects induces a transient interhemispheric coherence in the alpha-band; while such interhemispheric synchronization was absent during the presentation of meaningless objects or passive viewing. In the present study, Das et al. assessed interhemispheric coherence during midline object recognition using EEG, and they evaluated also the interhemispheric functional connectivity, after EEG source reconstruction. The Authors replicated the alpha desynchronization effect from the 2001 study, but not the interhemispheric coherence results. Nonetheless, they found modulations in the imaginary part of coherency and Granger causality estimates between left and right occipito-temporal regions.

I agree that the use of more advanced methods and connectivity measures is interesting to investigate object recognition and it may help to provide novel insights into this neural mechanism, and the Authors should be applauded for pursuing this path. However, I have some major concerns that make the current work not suitable for publication in its current form. My concerns are listed below in order of occurrence. I hope that these points will help the Authors to improve their work and manuscript. 

We thank the reviewer for the valuable comments and suggestions. We have made all the necessary changes, which we believe has improved the quality of our manuscript. We hope this makes our manuscript suitable for publication in PLoS One.

* It is unclear whether the imaginary coherency effects are also specific for the alpha-band. The Authors should explicitly mention that in the Abstract.

We thank the reviewer for pointing this out. Yes, the effects on imaginary coherence were specific for the alpha band. We have now clarified this in the third paragraph of the abstract. 

* The Authors use interchangeably the words “synchronization” and “synchronisation”. Please be consistent. In general, a final proofreading is needed to correct typos and make the grammar more consistent.

All mentions of ‘synchronization’ have been changed to ‘synchronisation’, throughout the manuscript.

* In the first paragraph, the Authors introduce the term “alpha rhythm” without specifying the frequency range. These details are important, especially for a broader, less expert readership.

We agree with the reviewer that it is important to specify the frequency range of the alpha rhythm. We have now included the frequency range 8-12 Hz within parenthesis, at the first mention of alpha rhythm in the introduction. 

Also, the use of parentheses to report time windows is confusing. I suggest to avoid that, and instead report simply the extremes of each time interval using an en dash (e.g., “117–373 ms”).

We have changed all time windows to the suggested format (removed parenthesis and used en dash). 

* Mima et al. employed familiar and unfamiliar/meaningless objects. I am not sure what “non-object stimuli” are. I recommend the Authors to clarify the design of the previous study and report its results using unambiguous terminology. This will also help the reader to understand the rationale and details of the current work.

We apologise for the confusion in terminology. We have now changed all mentions of ‘non-object’ to ‘meaningless object’ throughout the manuscript. In the introduction paragraph, we changed the word from ‘non-object’ to ‘unfamiliar/meaningless object’. Familiar objects are still referred to as ‘Objects’. 

* (minor) Figure 1. More or less 50% of this figure is occupied by the electrodes layout (1A), which does not really add any useful information and it is not central for the description of the procedure. More space should be given to the design of the experiment and the stimuli (1B-C).

Sizes have now been adjusted to make figure 1A smaller than the other two figures.

* The Authors mentioned that anatomical MRI images “were either MPRAGE or 3D-T1 images from a 3T scanner”. What does that mean? Did the Authors used different sequences for anatomical image acquisition across subjects? Furthermore, several technical details are missing. What parameters were used for data acquisition? What is the model and manufacturer of the scanner?

The nomenclature used by different scanner types (Siemens vs Philips) to collect anatomical images is different – MPRAGE or 3D-T1 respectively. Depending on the scanner type, the name of the sequence differs. These are standard sequences offered by two the scanner types. Including complete details of the sequences may be beyond the scope of this study and hence we have not mentioned it. We now state that “T1-weighted anatomical magnetic resonance images (MRI) were obtained for all participants from a 3T scanner.”

* (minor) Sensor data analysis. I suggest to change the name of the subsection to “Sensor-space analyses”.

We have changed the title of the subsection to sensor space analysis as suggested. 

* Sensor data analysis. For one subject in one session, one channel was rejected and interpolated. However, no motivation for doing that has been provided. Was the channel contaminated by excessive artifacts during the recording? If that is the case, how were these identified? Please motivate this choice.

We have addressed this in the first paragraph under ‘Sensor-space analyses’. “For one of the participant’s data from one block, the signal from channel CPz was intermittently saturated and had large fluctuations. Hence, CPz channel was removed from this particular dataset and later interpolated using EEGLAB toolbox.”

* (minor) Sensor data analysis. The Authors report mean and standard deviation across subjects of the number of trials used for the successive analyses. In addition to the SD, the range (minimum-maximum across subjects) may actually provide a more informative measure of the amount of trials surviving preprocessing.

We have now included the maximum and minimum number of trials for each category (objects and meaningless objects). “An average of 44 ± 4 (mean ± SD; maximum 49 and minimum 36) meaningless objects trials and 44 ± 3 (mean ± SD; maximum 49 and minimum 37) object trials per participant were finally used for the analyses.”

In the first paragraph under Results section, the sentence has been changed to “An average of 89 ± 6 (mean ± SD, maximum 98 and minimum 78) trials (including object and meaningless object trials) per participant were used for final analysis.”

* Sensor data analysis. The Authors should specify the frequency range for the estimation of alpha-band coherence. This information is currently missing in the Methods section.

We have edited the 3rd paragraph under Sensor space analyses of the Methods section, to include the frequency range 8-12 Hz.

* Using the imaginary part of coherency reduces the possible confounding effects of zero-lag interactions. However, when estimated in sensor-space, even this measure does not provide any meaningful interpretations in terms of underlying cortical sources, but it provides only an estimate of statistical dependencies between sensors (see Brunner et al., Front. Comput. Neurosci., 2016; Van de Steen et al., Brain Topogr., 2016). To avoid any confusion, the authors should clarify that. 

We have now included this point under the discussion section with the appropriate references, as suggested. 

* Source localization. How was the noise covariance matrix estimated? Which time window have the Authors employed for estimation?

We have addressed this point in the 2nd paragraph under source data analysis and have included the time window (-2 to -0.001s). 

* Source localization. The Authors mentioned that Granger causality estimates were derived to assess the interactions between left and right ROIs. It is unclear what variant of Granger causality was used: time-domain definition, Geweke’s frequency-domain definition, or else? References are also missing. Without these details, it is impossible to assess the quality of the methodological procedures.

We have now included the details of the Granger causality method used. “Spectral Granger causality values were computed in the frequency domain for both left ROI-to-right ROI and right ROI-to-left ROI directions, using FieldTrip toolbox. ” 

* Statistical analysis: Sensor space. It is unclear why the Authors restricted their analysis to the alpha-band. I understand the goal to replicate the previous findings, but thanks to modern computers and cluster-based permutation tests for statistical comparison (e.g., see Maris and Oostenveld, J. Neurosci. Methods, 2007), a full time-frequency analysis in sensor-space is called for.

The reviewer is absolutely right in saying that our motivation was to replicate the results of Mima et al. and therefore their results determined our frequency of interest, however, a data-driven approach may be very useful. We had in fact computed time frequency power plots for frequency range of 2-50 Hz as shown in figure 2. However, we found a significant decrease in power only in the alpha band, as reported in the previous study. Hence, our subsequent analyses were restricted to the alpha band. 

We performed a cluster based permutation analysis for the alpha band across all channels and time points from 0 to 1000ms.

We have further elaborated on this in another comment# below.

* Figure 2. The font of axes and colorbar ticks’ labels should be increased. Units are missing for the labels of the x and y axis. Please use the letters A-B to distinguish between the left and right panel in the figure. I also recommend to center the scale of the colorbar around zero (i.e., no increase/decrease) to improve interpretability of the results.

We have now edited Figure 2 as per the reviewer’s suggestions.

* Figure 3. In addition to describing the difference in bars’ texture between the two types of stimuli (solid black/checked) in the caption, please add a legend showing such difference. This information should be immediately clear from the figure. Furthermore, I suggest to rename the label of x-axis to “Preselected time windows” or something like that.

We have made the recommended changes to the figure.

* Figure 4. Same comments as for Figure 3 (see above).

We have made the recommended changes to the figure.

* Sensor-space coherence/iCoh results. Something unclear to me is the reason why the Authors decide to report and comment only the statistical comparison with respect to baseline if it is not the best at capturing the main results that they are trying to convey (i.e., differences between familiar and unfamiliar objects). The Authors did such comparison only for the power (see Figure 5). Unless the Authors have a specific reason to avoid that, I suggest adding to the statistical design the comparison familiar vs. unfamiliar/meaningless objects. This analysis could be done in the entire time-frequency space, by using the nonparametric cluster-based permutation approach (as I mentioned in a previous point).

* Sensor-space results. Like for the other analyses, it is unclear why the Authors restricted the comparison in few time windows and one frequency band.

#As mentioned in an earlier comment, we evaluated power spectral changes for frequency range of 2 – 50 Hz as shown in Figure 2. However, we saw significant decrease in power only for the alpha frequency range. This was also supported by the results of the previous study by Mima et al. So, we restricted our subsequent analyses to the alpha band. The results of initial analyses shown in Figures 3 and 4 were performed to replicate results from the Mima et al. study. 

Exactly as the reviewer suggested, we further used a data-driven approach (cluster-based permutation analysis) to spatially and temporally localize significant changes in power and coherence. For this, we used data from all 64 channels and time points from 0 to 1000ms following the visual stimulus, which is a pretty wide time window to study cortical events relevant to object recognition. A frequency resolution of 1Hz restricted our temporal resolution to 100ms. We found significant decrease in alpha power at 600 and 700ms as shown in Figure 5. For better depiction of results in the form of a bar chart, we averaged the data to 250ms time windows (figure 5B). 

For the source analysis, the number of vertices obtained were 5124. To reduce the number of multiple comparisons without compromising on spatial resolution, we had to pool the data to distinct time windows of 250ms to achieve the same frequency resolution of 1Hz as in sensor space analysis.

For alpha power, we focused on the time window of 500-750ms that showed significant difference in the permutation analysis in sensor space (600 & 700ms). 

For interhemispheric functional connectivity, we calculated imaginary coherence for time windows of 250ms from 0 to 1000ms and found a significant difference in only in the first time window (0-250ms) as shown in figure 8. Hence, for further estimation of granger causality, we restricted ourselves to this time window (Figure 9). 

In summary, for the sensor space, using a data-driven approach, we could achieve a temporal resolution of 100ms and frequency resolution of 1Hz. For the source analysis, to achieve the same frequency resolution along with improved spatial resolution, we had to compromise slightly on the temporal resolution and decrease it to 250ms instead of 100ms. Yet, we were successful in extracting the significant differences in connectivity. 

* Also, the Authors mention that “In order to get a more conservative and robust estimation of the coherence between the two regions, imaginary part of coherency was computed.” This sentence is misleading and should be rephrased. Coherence and iCoh measure essentially two different things. 

We have rephrased the sentence – “In order to minimize volume conduction effects that may give rise to spurious connectivity between regions, imaginary part of coherency was computed (Nolte et al., 2014)”

* Figure 8. Same comments as for Figure 3 (see above).

We have made the recommended changes to the figure.

* Granger causality results. Without specifying how Granger causality was estimated it is impossible to assess the reliability of the results shown.

We have now included details of the method for estimation of Granger causality.

* I agree that the advantage of using EEG source reconstruction is its high temporal resolution, combined with a reasonably acceptable localization of cortical sources of activity. However, the Authors did not take full advantage of this characteristics of the EEG signals, by restricting their analyses to four windows of interest.

We agree with the reviewer in that we have not fully exploited the advantage of high temporal resolution of the technique. However, we have been able to achieve precise localization of activity and narrow down our findings to a temporal resolution of 250ms. We had to pool our data to 250ms time windows to achieve the frequency resolution of 1 Hz. Future studies can focus on further improving upon the current methodology by using MEG to decrease volume conduction effects or increasing the temporal resolution of the effects.

We have mentioned this under the discussion section.

* What is the definition of “true functional connectivity” for the Authors? I suggest to expand on that. And, in general, I recommend to be a bit more cautious in drawing certain conclusions.

By true functional connectivity, we meant functional connectivity that excludes self-correlations arising from volume conduction as described by Nolte et al., 2014 and Haufe et al., 2013. We have rephrased this sentence in the discussion section.

---

## [Editor Report · Decision Letter 1]

22 Jun 2022

Evaluating interhemispheric connectivity during midline object recognition using EEG

PONE-D-21-26846R1

Dear Dr. Thirugnanasambandam,

We’re pleased to inform you that your manuscript has been judged scientifically suitable for publication and will be formally accepted for publication once it meets all outstanding technical requirements.

Kind regards,

Kiyoshi Nakahara, PhD

Academic Editor

PLOS ONE

Additional Editor Comments (optional):

I apologize for the delay in finalizing our decision; we received your revised manuscript on April 11 and asked the previous reviewer to review it, but have not received a reply. Therefore, I have reviewed the revised manuscript and have decided to accept for publication.

---

## [Editor Report · Acceptance letter]

17 Aug 2022

PONE-D-21-26846R1 

Evaluating interhemispheric connectivity during midline object recognition using EEG 

Dear Dr. Thirugnanasambandam:

I'm pleased to inform you that your manuscript has been deemed suitable for publication in PLOS ONE. Congratulations! Your manuscript is now with our production department. 

Kind regards, 

on behalf of

Dr. Kiyoshi Nakahara 

Academic Editor

PLOS ONE